# Continuous Production of Fumaric Acid with Immobilised *Rhizopus oryzae*: The Role of pH and Urea Addition

Reuben Marc Swart [ID], Dominic Kibet Ronoh [ID], Hendrik Brink [ID] and Willie Nicol *[ID]

Department of Chemical Engineering, University of Pretoria, Lynnwood Road, Hatfield, Pretoria 0002, South Africa; reuben.swart@tuks.co.za (R.M.S.); u16278072@tuks.co.za (D.K.R.); deon.brink@up.ac.za (H.B.)
* Correspondence: willie.nicol@up.ac.za

**Abstract:** Fumaric acid is widely used in the food and beverage, pharmaceutical and polyester resin industries. *Rhizopus oryzae* is the most successful microorganism at excreting fumaric acid compared to all known natural and genetically modified organisms. It has previously been discovered that careful control of the glucose feed rate can eliminate the by-product formation of ethanol. Two key parameters affecting fumaric acid excretion were identified, namely the medium pH and the urea feed rate. A continuous fermentation with immobilised *R. oryzae* was utilised to determine the effect of these parameters. It was found that the selectivity for fumaric acid production increased at high glucose consumption rates for a pH of 4, different from the trend for pH 5 and 6, achieving a yield of $0.93 \, \text{g g}^{-1}$. This yield is higher than previously reported in the literature. Varying the urea feed rate to $0.255 \, \text{mg L}^{-1} \, \text{h}^{-1}$ improved the yield of fumaric acid but experienced a lower glucose uptake rate compared to higher urea feed rates. An optimum region has been found for fumaric acid production at pH 4, a urea feed rate of $0.625 \, \text{mg L}^{-1} \, \text{h}^{-1}$ and a glucose feed rate of $0.329 \, \text{g L}^{-1} \, \text{h}^{-1}$.

**Keywords:** fumaric acid; *Rhizopus oryzae*; immobilised reactor; crabtree effect; urea addition; urea cycle; pH

## 1. Introduction

The global market is experiencing a shift towards more renewable feedstocks and procedures. This is largely motivated by the drive to find alternatives to crude oil-derived chemicals. Focus has been brought onto the production of bio-based chemicals from renewable carbohydrate sources [1,2]. Dicarboxylic acids are one group that have been identified to be ideal bio-based platform chemicals. Fumaric acid, malic acid and succinic acid were highlighted as they are produced in the tricarboxylic acid (TCA) cycle. However, they are not readily excreted by many organisms in large quantities.

Fumaric acid (FA) has a global market size of 660.9 million USD and is expected to grow at a rate of 5.5% annually for the period 2021–2026 [3]. FA has great potential to feed the 2.77 billion USD petrochemically derived maleic anhydride market [4]. Using a high-yielding dehydration process, FA can be converted to maleic anhydride [5]. FA is largely used in the food and beverage industry for pH control and as a flavour enhancer. Other industries, including paper resin, unsaturated polyester resin, animal feed and pharmaceutical, also use FA [6–8]. FA added to animal feed has been found to reduce methane production by 32% [9]. Fumaric acid esters have been found to be an effective treatment for multiple sclerosis and psoriasis [10,11].

Frits Went and Hendrik Coenraad Prinsen Geerligs first discovered *Rhizopus oryzae* ATCC 20344 in 1895. Although it has since been correctly renamed *Rhizopus delemar*, it is still referred to as *Rhizopus oryzae* in most of the literature [12]. Fumaric acid production by *R. oryzae* has not been matched by any other natural organism or by a genetically modified strain of *Saccharomyces cerevisiae* or *Escherichia coli* [6]. The production of fumaric acid by

*R. oryzae* is sensitive to a variety of environmental conditions, the most influential being morphology, pH, nitrogen availability and metal ion concentrations [13–16].

An industrially viable process will require a continuously steady high-yielding rate of production. It was found that a continuous low urea (nitrogen source) feed during the production phase prolonged the fermentation and resulted in steady FA production [17]. This strategy was tested together with the glucose-limited feed strategy and it was found to be successful in eliminating ethanol production while maintaining a steady production rate [18]. Light has been shed on the mechanism by which fumarate is produced. The urea cycle has been found to also contribute to the production of fumarate during nitrogen starvation conditions [15]. This may be the key to why FA production is highly dependent on the nitrogen concentration in the medium. The effect of varying the urea feed rate has been previously tested under excess glucose conditions [17]. Further experimental investigation is warranted to better understand the interplay between the glucose uptake rate and the urea feed rate.

The pH of the medium is known to have an effect on the metabolism and morphology of *R. oryzae*. Investigations as to the effect of pH on FA production have been focused on cell morphology and production in batch fermentations [16,19,20]. A thermodynamic analysis of dicarboxylic acid production highlights the energy cost of acid transport into the extracellular medium [21]. Dicarboxylic acids do not diffuse across the cell wall like ethanol, and therefore, require a transporter to overcome the concentration gradient over the cell wall. The cost of transporting fumaric acid out of the cell is a function of both the pH and the extracellular concentration. Transport cost is inversely proportional to the pH and proportional to the extracellular concentration. This is counter to the assumption that the production of FA is adenosine triphosphate (ATP) neutral [22]. The implication of this is that more glucose is used in the production of FA since ATP needs to be produced for export costs. The role of pH on FA production with *R. oryzae* has previously been investigated but only in excess glucose batch fermentations [16,19].

The current study builds on previous work by this research group [18,23,24]. In the previous work, a novel bio-reactor was developed in which *R. oryzae* was immobilised while allowing careful control of the fermentation conditions in the bio-reactor. It was previously found that ethanol formation can be minimised by carefully controlling the glucose feed rate to the bio-reactor. However, the increased fumaric acid's effect on glucose yield ($0.802 \, \mathrm{g \, g^{-1}}$) was still markedly lower than the theoretical maximum yield of $0.97 \, \mathrm{g \, g^{-1}}$ [21]. The current study aims to utilise the novel bioreactor and the glucose-controlled feed strategy to further investigate medium conditions that influence the yield and production rate of fumaric acid.

## 2. Results and Discussion

### 2.1. Experimental Window Determination and Repeatability

All of the fermentations were conducted with immobilised *R. oryzae* in a bio-reactor, which allowed for the control of the fermentation conditions. These primarily included the pH, temperature, dilution rate and medium composition. An experimental run consisting of around 200 h remained at constant pH and urea addition rate while the glucose feed rate was altered. Different glucose rates during a run represent processing windows. Each window was considered as a separate processing condition where distinct rates and yields could be calculated. The window spans typically lasted 36 h, and the last 24 h were considered for the calculation of the specific state. Figure 1 indicates the glucose feed rate and NaOH dosing rate for four separate windows; it is evident from the figure that rapid dynamic changes occurred around the step while more stable dosing rates occurred towards the end of the window. In order to calculate the true production rate of metabolites, a mole balance was performed over the last 24 h of operation where differences in high-performance liquid chromatography (HPLC) determined the concentration values (shown in Figure 1a), and the continuous dilution rate was used (see Section 3.5). The procedure enabled four to five separate processing conditions within a single run. The calculated fumaric acid production rates (black squares) can be seen in Figure 1b for each window.

The consolidated data from each window will subsequently be used for the analysis of the fermentations.

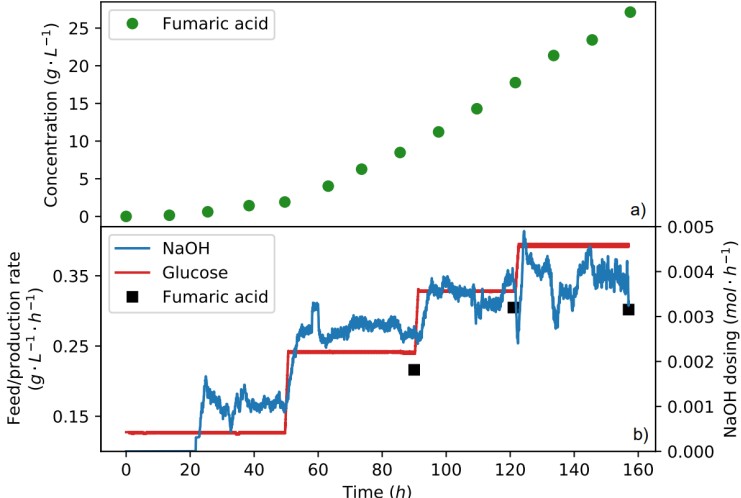

**Figure 1.** An illustration of how stable acid production was determined and how concentration profiles were used to determine volumetric rates of production. (**a**) The HPLC fumaric acid concentration profile taken from the reactor. This is data from a single fermentation. (**b**) A comparison between the glucose feed rate and the rate of NaOH dosed to maintain the pH. The fumaric acid production rates calculated from the HPLC concentrations and the dilution rate are also shown (black squares).

It is imperative to analyse the repeatability of the operation at the same conditions. Windows can be repeated within an experimental run (at least 75 h apart) or in completely separate runs. In Table 1, four repeat windows are compared at different glucose-addition rates. Some are from the same run while others are from separate runs, as indicated in the table. In the table, it can be seen that the average difference between the fumaric acid production rates is 3.5%, with the average difference between the glucose consumption rates being 2.7%. These repeatable results indicate that the activity of the biomass remained stable over the fermentation for different pH values and urea feed rates. This accordingly enables comparison between different windows regardless of the time of the fermentation.

**Table 1.** Repeatability and stability of fumaric acid production evaluated through glucose consumption and distribution.

| | Separate Run Repeats | | Repeats within a Run | |
|---|---|---|---|---|
| Glucose feed rate (g L$^{-1}$ h$^{-1}$) | 0.132 * | 0.197 * | 0.227 * | 0.263 † |
| Glucose consumed (g L$^{-1}$ h$^{-1}$) | 0.133 0.142 | 0.197 0.195 | 0.222 0.226 | 0.240 0.236 |
| Fumaric yield (g g$^{-1}$) | 0.733 0.774 | 0.727 0.695 | 0.900 0.894 | 0.943 0.912 |
| By-product yield (g g$^{-1}$) | 0.076 0.073 | 0.074 0.040 | 0.093 0.116 | 0.139 0.108 |

* Experimental conditions: pH 5 and urea feed rate of 0.625 mg L$^{-1}$ h$^{-1}$; † Experimental conditions: pH 4 and urea feed rate of 0.255 mg L$^{-1}$ h$^{-1}$.

## 2.2. The Role of pH and Urea Feed Rate on the Production of Fumaric Acid

Figure 2a–c are parity plots between the glucose fed and the glucose consumed for various conditions. At low glucose feed rates, it can be seen that all the glucose feed is consumed. The $y = x$ line (black line) indicates the point of full glucose consumption. Points below the line indicate glucose accumulation and suggest suboptimal glycolytic flux. For the urea feed rate of 0.625 mg L$^{-1}$ h$^{-1}$ and pH 4, it can be seen that there was full consumption of glucose at all the glucose feed rates. For pH 5, at the highest glucose feed rate of 0.329 g L$^{-1}$ h$^{-1}$, a slight drop in glucose consumption can be seen. However, looking at the glucose consumption for pH 6, from a glucose feed of 0.197 g L$^{-1}$ h$^{-1}$, glucose

accumulation is observed. It is clear that at a lower pH, more glucose can be consumed. This indicates that the pH inhibits some crucial pathway in the consumption of glucose, either in the uptake of glucose or in the metabolism of it. The rate at which glucose is consumed is to be maximised, provided the fumaric acid yield is maintained, as this will increase fumaric acid productivity, which is of key importance. It was also found that the urea feed rate affected the glycolytic flux. The lowest urea feed rate of $0.255 \, mg \, L^{-1} \, h^{-1}$ has a negative effect on the glycolytic flux. It can be seen that for both pHs 4 and 5, the glycolytic limit is reached first for the lowest urea feed rate. Through the repeatability experiments, it was determined that, at least within the time frame of the experiments conducted (approximately 200 h), no decay of the glucose consumption rate was seen for the urea feeds of 0.255 or $0.625 \, mg \, L^{-1} \, h^{-1}$. This shows that the decreased glycolytic flux seen for the lowest urea feed is a result of the condition and not the length of the experiment.

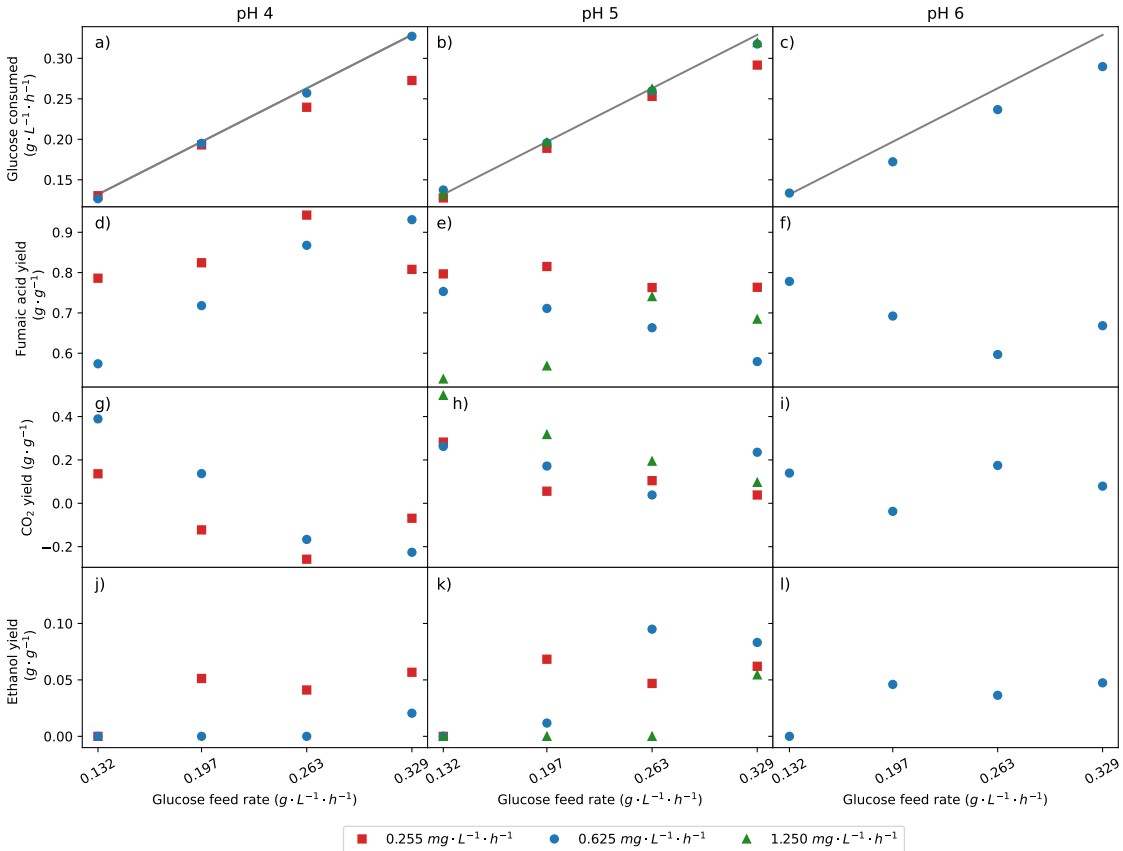

**Figure 2.** The effect of pH and the urea addition rate on glucose consumption and distribution. (**a**–**c**) A parity plot between the glucose feed and glucose consumed. (**d**–**f**) The yield of fumaric acid from glucose. (**g**–**i**) The yield of $CO_2$ from glucose. (**j**–**l**) The yield of ethanol from glucose. As indicated in Table 1, repeat experiments were conducted, and the average values of the two experiments are presented. For the conditions that were not repeated, single fermentation values were used.

Clear improvement of fumaric acid production has been found by altering the pH. Comparing the yields of fumaric acid achieved for the different pHs tested (Figure 2d–f), it can be seen that there is a difference in the trends as the glucose feed rate is increased. For pHs 5 and 6, the yield decreased as the glucose feed rate is increased, but the yield for pH 4 increased. The yield of $0.94 \, g \, g^{-1}$ fumaric acid on glucose achieved for pH 4 at a urea feed of $0.255 \, mg \, L^{-1} \, h^{-1}$ is the highest reported in the known literature [6]. This is closely approaching the theoretical maximum of $0.97 \, g \, g^{-1}$ [21]. The yield achieved at the higher urea feed rate of $0.625 \, mg \, L^{-1} \, h^{-1}$ is $0.93 \, g \, g^{-1}$, this has the massive benefit of a higher glycolytic flux, consuming all of the glucose feed. Thereby achieving productivity

of $0.305\,\mathrm{g\,L^{-1}\,h^{-1}}$ fumaric acid. Total consumption of glucose will allow for the extended production of fumaric acid since there will be no problematic accumulation, and downstream processing is simplified. Comparing the yields of $CO_2$ shown in Figure 2g,h,l), it is seen that the highest fumaric acid yields correspond to the lowest $CO_2$ yields. At these points, the $CO_2$ yields are negative, indicating that $CO_2$ is being consumed by pyruvate carboxylase (EC 6.4.1.1) in the pathway of fumaric acid production. This is a result of the high yield of fumaric acid production but also suggests that less carbon is being directed to the TCA cycle.

A higher urea feed rate is shown to increase the glycolytic flux; however, the effect on the distribution of glucose must also be taken into account. It can be seen for both pHs 4 and 5 that the highest yield of fumaric acid is achieved consistently for the lowest urea feed. This confirms the view that urea addition inhibits the production rate of fumaric acid but shows that there is a greater influence on the yield of fumaric acid. It can clearly be seen that as the feed rate of urea is increased, the yield of fumaric acid decreases. Nitrogen availability influences the glucose uptake rate and how the glucose consumed is metabolised. Looking at the fumaric acid yields achieved for the urea feed of $1.25\,\mathrm{mg\,L^{-1}\,h^{-1}}$, it can be seen for the first two glucose feed rates that the above trend is observed; however, it is not followed for the higher glucose feed rates. There is a clear trade-off when it comes to the effect of urea feed rate on fumaric acid production. On the one side, the glycolytic flux is inhibited by low urea addition, while on the other side, the fumaric acid yield is clearly higher at low urea feeds. Since fumaric acid production will be a key optimisation parameter, a balance between these two effects will result in optimum production.

The point at which glucose accumulation begins is consistently preceded by ethanol production. Ethanol is the major by-product and all other by-products follow a similar trend to ethanol. Figure 2j–l shows the ethanol yields for the various conditions. If one compares the glucose feed rate where glucose accumulation begins, it can be seen that at that feed rate or the preceding feed rate, ethanol production began. This ties into the function of glycolysis; once the carbon sinks of fumaric acid production and the TCA cycle are at capacity, ethanol overflow begins. Increasing the glucose feed rate further leads to glucose accumulation, as no more glucose can be accommodated through glycolysis. Comparing the ethanol yields for the three pHs at the urea feed of $0.625\,\mathrm{mg\,L^{-1}\,h^{-1}}$, it can be seen that the pH clearly shifts the point at which ethanol breakthrough occurs. The ethanol breakthrough point is also influenced by the urea feed rate. At pH 5 it can be seen that as the urea feed rate was increased, the glucose feed rate at which ethanol breakthrough occurred increased. This clearly shows that an increased nitrogen feed rate increases the glycolytic flux that can be accommodated before ethanol breakthrough occurs. It is also evident that the yield of fumaric acid is unaffected by ethanol breakthrough.

The yields of $CO_2$ (Figure 2g–i) show why the yield of fumaric acid is unaffected by ethanol breakthrough. The increase in fumaric acid yield always corresponds to a decrease in the yield of $CO_2$. The yield of $CO_2$ gives insight into ATP production since $CO_2$ is largely produced through the TCA cycle. Using the metabolic flux model described in Section 3.6, the fraction of carbon consumed that is directed to the TCA cycle was calculated (Figure 3a,b). As the yield of fumaric acid increases, the fraction of carbon that is directed to the TCA cycle decreases and this counteracts ethanol breakthrough, maintaining the yield of fumaric acid.

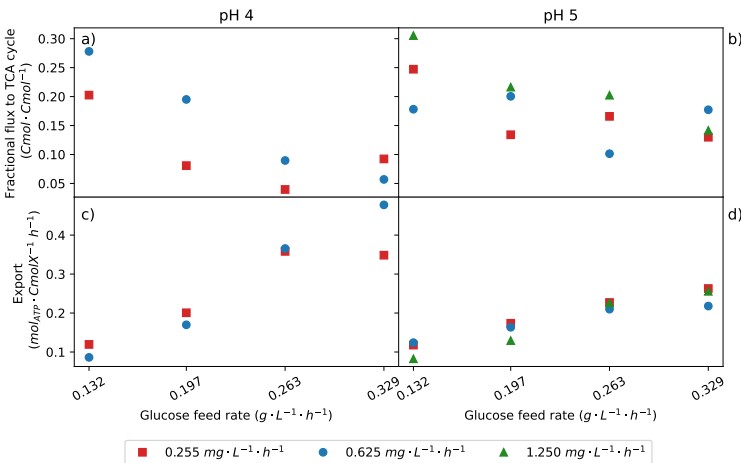

**Figure 3.** The effect of the urea addition rate on the energy parameters at pHs 4 and 5 (**a**,**b**). The fractional amount of glucose that is directed to the TCA cycle (**c**,**d**). The ATP required to export the acids produced into the medium. As indicated in Table 1, repeat experiments were conducted, and the average values of the two experiments are presented. For the conditions that were not repeated, single fermentation values were used.

It was found that there was a mass increase of biomass in the reactor during the production of fumaric acid. Figure 4 shows the yields of biomass on glucose consumed for the various conditions. It can be seen that as the urea feed rate is increased, the yield of biomass increases between the low and medium urea feed rates. This trend is not followed for the highest urea feed rate since a higher biomass yield is expected. Looking at the nitrogen to carbon ratios shown in Table 2, it can be seen that the nitrogen contents of the biomass produced during the production runs are less than that produced during the growth phase. The biomass initially grown is in a nitrogen excess medium. The nitrogen-starved environment had a clear influence on the biomass produced and is not equivalent to the biomass initially grown. The nitrogen contents in the final collected biomass were compared to the amount of nitrogen fed during the fumaric acid production phase, and it was found that all the nitrogen was accounted for. This suggests that all urea was absorbed during production and that higher urea feed rates resulted in biomass with more accumulated nitrogen. The total biomass increase over the fumaric acid production span was between 1.95 and 5.72 times the initial amount of biomass (obtained from the growth phase). This figure does not correspond to the total nitrogen increase in biomass over the production period that was between 41% and 119% (nitrogen accumulated over production divided by nitrogen present in biomass after growth) for the low and medium urea runs, and 247% for the high urea runs. The major difference between nitrogen and total biomass increase can be attributed to the accumulation of carbohydrates, as indicated in the flux model presented in Section 3.6. The nitrogen increase does, however, suggest that more proteins were synthesised during production, and it can be anticipated that this will increase the overall activity of the biomass. Activity increases were, however, not observed when considering the stability of the repeatability data. This clearly suggests that fractions of the biomass became inactive during production as biomass steadily grew due to urea addition. The relative stability of the biomass activity further hints that low urea addition resulted in less biomass death than high urea addition. Enhanced usage of the urea cycle at low urea feed rates provides a plausible reason for the slower rate of biomass decay at these conditions.

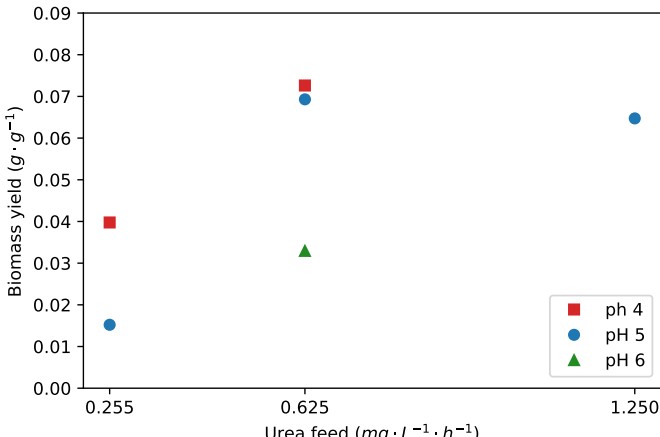

**Figure 4.** The effect of pH and urea addition on the yield of biomass on glucose. As indicated in Table 1, repeat experiments were conducted, and the average values of the two experiments are presented. For the conditions that were not repeated, single fermentation values were used.

**Table 2.** Experimental parameters analysing the yield and elemental composition of *R. oryzae* biomass.

| Phase | pH | Urea Feed ($mg\,L^{-1}h^{-1}$) | $Ygx\,(g\,g^{-1})$ | H:C | N:C |
|---|---|---|---|---|---|
| Growth | 5 | Excess | 0.180 | 1.410 | 0.189 |
| Production | 5 | 0.255 | 0.013 | 0.807 | 0.128 |
| Production | 5 | 0.625 | 0.080 | 0.644 | 0.120 |
| Production | 5 | 1.250 | 0.066 | 1.041 | 0.148 |
| Production | 4 | 0.625 | 0.071 | 1.030 | 0.123 |

It has been seen at low pH (pH 4) and especially at low nitrogen (urea feed of 0.255 mg $L^{-1}$ $h^{-1}$) that there is a very low rate of respiration or carbon flux to the TCA cycle. This indicates that there is a lower rate of ATP production under these conditions. Since it is known that the export of fumaric acid comes at an additional ATP cost [21], it is expected that the export cost is lower under these conditions. Figure 3c,d shows that the export costs are, in fact, the highest under these conditions. This severe contradiction cannot be explained by using enhanced cycling of the urea cycle. The urea cycle is expected to be enhanced under low nitrogen feed conditions, where more ATP expenditure takes place in producing fumarate [15,25]. The extremely low flux to respiration under these conditions is counterintuitive when considering the enhanced costs of fumaric acid export, as well as the enhanced maintenance cost of the urea cycle. A plausible explanation is that the efficiency of generating ATP in oxidative phosphorylation is severely enhanced under the low pH and nitrogen conditions. This postulate was tested in the flux model, where the oxidative phosphorylation values(P/O) higher than three were obtained to close the energy balance. Since the theoretical maximum for P/O values is three, the observed contradiction could not be elucidated. Accordingly, it is highly likely that the calculated export costs do not represent the functioning of the organism.

Dicarboxylic acids can be present as solid undissociated acid, aqueous undissociated acid and then dissolved in a dissociated form, of which there is a first and second conjugated base. The equilibrium ratio between these forms is dependent on the molar concentration of the acid, temperature and pH. At pH 6 most of the acid will be present, as the second dissociated form requires the most alkali to be added if the pH is to be maintained. At low pH values, the acid becomes undissociated, and depending on the specific acid, solid acid can form. Solids will begin to form for fumaric acid at a pH below 4.11 and a temperature of 25 °C [21]. Assuming an industrial production concentration of 1 mol $L^{-1}$ fumaric acid, operation at a ph of 6 would require 2.97 times the amount of NaOH than would be required at a pH of 4. This is an attractive operating condition since it results in the double benefit of less alkali isneeded to maintain the pH and less acid used in downstream re-acidification

of the medium; thus, decreasing the overall cost of fumaric acid production. Together with the results presented for pH 4 it can be seen that this is the ideal condition for fumaric acid production.

The pH also has an effect on the ionic and osmotic stress on the cell. The osmotic and ionic strength increase from approximately $0 \, \text{mol} \, L^{-1}$ at a pH of 2 in an S-curve shape to the maximum of $3 \, \text{mol} \, L^{-1}$ at a pH of 6. Operation at pH 4 results in an osmotic pressure that is a third of that at pH 6 [21]. These two factors are important since such stresses have been found to result in the production of unwanted by-products [26]. Through analysis of the pH 6 experiments, it was determined that pH 6 provided unfavourable fumaric acid yields, glucose accumulation and experienced high levels of ethanol production, as shown in Figure 2c,f,i, respectively. In Figure 4, it can also be seen that the production of biomass was disrupted by the high pH. This confirms the literature that suggested high osmotic and ionic stress caused by a high pH favoured by-product production. pH 6 is clearly an unfavourable condition for fumaric acid production.

## 3. Materials and Methods

### 3.1. Microorganism and Culture Conditions

*Rhizopus oryzae* (ATCC 20344) was used for all the fermentations of this study. The culture was as described by Naude and Nicol [23]. A spore concentration of $8 \times 10^6 \, \text{mL}^{-1}$ in sterile distilled water was made, and $10 \, \text{mL}$ were injected aseptically into each of the batch growth fermentations as the inoculum. The reactor was filled with the growth medium to the capacity of $1.08 \, \text{L}$.

### 3.2. Medium

All the fermentations used the same mineral medium with the urea, and glucose concentrations varied depending on the experiment. The mineral medium contained (all of the following values have units of $\text{g} \, L^{-1}$): 0.6 $KH_2PO_4$, 0.507 $MgSO_4 \cdot 7 \, H_2O$, 0.0176 $ZnSO_4 \cdot 7 \, H_2O$ and 0.0005 $FeSO_4 \cdot 7 \, H_2O$ [16]. The biomass was grown under batch conditions with $3.1 \, \text{g} \, L^{-1}$ glucose and $2.0 \, \text{g} \, L^{-1}$ urea [23]. The continuous production fermentations began with only the mineral solution. Urea was fed at a rate from $0.255 \, \text{mg} \, L^{-1} \, h^{-1}$ to $1.25 \, \text{mg} \, L^{-1} \, h^{-1}$, and glucose was fed at a rate from $0.132 \, \text{g} \, L^{-1} \, h^{-1}$ to $0.329 \, \text{g} \, L^{-1} \, h^{-1}$. In order to achieve low dilution rates, high-concentration solutions of both glucose and urea were made with $342 \, \text{g} \, L^{-1}$ and $16 \, \text{g} \, L^{-1}$, respectively. The dilution rate for the continuous production fermentations varied from $0.0018 \, h^{-1}$ to $0.0027 \, h^{-1}$, taking into account the glucose and urea additions, as well as the NaOH dosing. The urea solution incorporated the mineral solution to ensure that the mineral composition in the reactor remained constant over the duration of the experimental run. All the solutions were sterilised at $121 \, °C$ for $60 \, \text{min}$. All chemicals used were obtained from Merck (Modderfontein, South Africa).

### 3.3. Fermenter Design and Operation

During the production phase of the experiment, the fermenter was operated as a continuous stirred tank fermenter (CSTR). The feed of liquid into the fermenter was comprised of a mineral solution, glucose solution and a NaOH dosing solution. The flow rate of the mineral solution was kept constant for the entire run, the glucose solution varied depending on the desired glucose feed rate, and the flow rate of the NaOH solution varied dependent on the production rate of acids in the fermenter. This resulted in a varied dilution rate as the fermenter was also maintained at a constant volume; therefore, the rate that liquid was fed into the fermenter had to be equal to the flow rate of the liquid removed from the fermenter at a given time. Further information on the reactor operation is described by Swart [18]. For all of the experimental runs, the temperature was constant at $35 \, °C$.

### 3.4. Analytical Methods

All the fermentation medium concentrations were analysed with a High-Performance Liquid Analyser. The method is fully described by Naude and Nicol [23]. The fermenta-

tions were sampled at 12 h intervals. The dry cell mass was determined at the end of all experimental runs. Biomass measurements were not possible between the growth and production phases. The same biomass growth procedure was used for all experimental runs. Growth runs were terminated to determine the dry cell mass after growth and before production. The biomass was removed from the polypropylene, washed with 1 L of distilled water and then filtered through 541, 110 mm Whatman filter paper. The filter paper and biomass were then dried at 70 °C for 48 h before being weighed. The carbon, hydrogen, nitrogen and sulphur content of the biomass was determined by an Elemental Analyser. The Thermo Scientific Flash 2000 Organic Elemental Analyzer (Waltham, MA, USA) was used. The method was as follows: the gas pressures were 250 kPa for He and 300 kPa for $O_2$, the flow rates were 140 mL min$^{-1}$ for He measurement, 100 mL min$^{-1}$ for He reference and 250 mL min$^{-1}$ for $O_2$. The reactor temperature was 950 °C, and the oven was at 65 °C. The analytical duration was 620 s, the oxygen injection delay was 5 s, and the sample delay was 12 s.

### 3.5. Production Rate Calculations

Since the fermenter was operated as a CSTR and, therefore, has a dilution rate, the concentration profiles could not be directly used to determine the production rates. To separate the production rates from the concentration profiles, Equation (1) was used [27]. This equation calculates the molar change of a species in the fermenter by accounting for the entry, exit and production or consumption of a species. The desired variable is $r_j$; therefore, the equation was reworked into Equation (2). $\frac{dN_j}{dt}$ was calculated using the concentration profiles obtained from the HPLC analysis. Equation (3) illustrates how $\frac{dN_j}{dt}$ was calculated. It was assumed that the differential molar change term and the difference molar change terms were approximate for the calculations. The concentrations in between sample values were interpolated to calculate the difference. Equation (1) was solved for using Euler integration with a time increment of 1 s, this was the same increment that all other online measurements were sampled at. The effluent volumetric flow rate, $Q_e$, was comprised of the volumetric feed rate and the volume that was sampled from the fermenter at specific times.

$$\frac{dN_j}{dt} = Q_f C_j^f - Q_e C_j + r_j V, \tag{1}$$

$$r_j = \frac{\left( \frac{dN_j}{dt} - Q_f C_j^f + Q_e C_j \right)}{V}, \tag{2}$$

$$\frac{dN_j}{dt} \approx \frac{\Delta N_j}{\Delta t} = \frac{\Delta C_j}{\Delta t} V, \tag{3}$$

All products were accounted for from the HPLC analysis, as well as the concentration of glucose in the medium. This allowed for a mass balance to be used to calculate the rates of $CO_2$ and $O_2$. The mass balance took in the rates of glucose consumption and the production of fumaric acid, malic acid, succinic acid, pyruvic acid, ethanol and glycerol. It was found that there was an accumulation of biomass weight over the production run. The biomass accumulation was attributed to the growth of nitrogen-containing biomass enabled by the constant feed of urea, and the additional biomass was attributed to carbohydrate storage [28]. The amount of nitrogen-containing biomass growth was determined by assuming that the biomass formula remains constant after the growth phase into production and that all the nitrogen present in the urea feed was converted to biomass. Glycogen was chosen as the carbohydrate storage molecule. The rate of glycogen was then calculated as the remaining unaccounted mass.

### 3.6. Metabolic Flux Model and Energy Calculations

In order to better understand the effect of the environmental changes on the physiology, a metabolic flux model was setup (this can be seen in Figure 5).

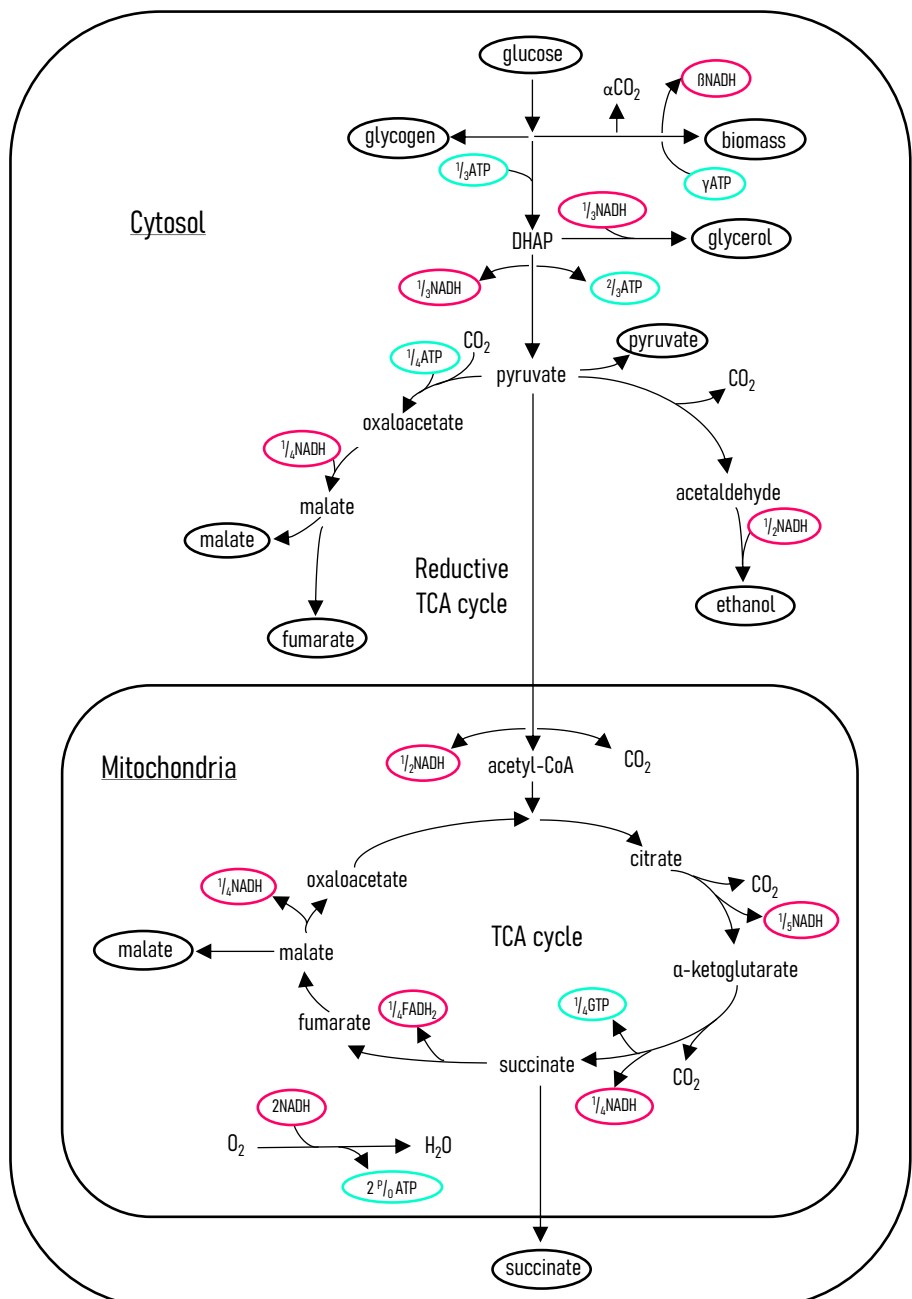

**Figure 5.** *R. oryzae* metabolic flux model. The flux model is written on the basis of carbon moles; this results in the fractional amounts of the energy-related compounds. The compounds that interact with the medium are circled in black.

The method followed is described by Villadsen [27]. The metabolic pathways included in the flux model described how glucose is distributed to biomass and all the other metabolites [15,22,29]. A lumped approach was used to account for the anabolism, which included accounting for $FADH_2$ and GTP as NADH and ATP, respectively [27]. The model allows for the calculation of energy production and consumption. A matrix was setup by using mass balances over each reaction and an NADH balance. The consumption and production rates of the glucose, biomass and the metabolites were used as specifications in the matrix. There were two additional specifications that were not needed to solve the matrix, namely $CO_2$ and $O_2$. These were used to confirm that the metabolic flux model agrees with the mass balance. Solving the flux of all the individual metabolic rates allowed for the ATP production and consumption rates to be calculated. The efficiency of oxidative

phosphorylation indicated by the P/O number has been assumed to be constant for all the conditions at a value of 1.5 moles of ATP per atom of oxygen consumed. This value is suggested by [27], and although it is unlikely that it remains constant, this assumption will be accounted for in the discussion of the results. Another assumption that needed to be made was the amount of energy used for the production of new biomass enabled by the urea feed. The value chosen is $2.5\,mol_{ATP}/Cmol_X$, since the system is known to be fully aerobic [18]. The amount of $CO_2$ released, $\alpha$, and the amount of NADH produced, $\beta$, per mole of biomass were calculated to be $0.116\,mol_{CO_2}\,Cmol_X^{-1}$ and $0.116\,mol_{NADH}\,Cmol_X^{-1}$, respectively. These values were calculated using a mass balance over the glucose to biomass equation and the biomass formula found from the elemental analysis.

The additional ATP cost of exporting fumaric acid and the other dicarboxylic acids into the medium needed to be determined. The literature outlines the equations and procedure to calculate the ATP cost of exporting dicarboxylic acids into the medium [21]. The ATP cost is dependent primarily on the medium pH and the concentration of the specific acid. Equations (4) and (5) are used to relate the medium pH to the ratio between the intracellular and extracellular acid concentrations for a specific transporter. There are three transporters that are ordinarily present, namely a uniport ($n = 1$), symport ($n = 0$) and antiport ($n = -1$). The proton motive force, $pmf$, and the intracellular, $A_i$, values were suggested to be $0.15\,V$ and $1 \times 10^{-3}\,mol\,L^{-1}$. It was found that the energy costs predicted were not sensitive to the variation of these parameters. The transporter used at a set of conditions is selected by the proximity of its achievable concentration ratio to the required minimum concentration ratio dictated by the medium. The ATP cost of each transporter varies as a result of the number of protons that need to be exported for each molecule of acid to balance the charge. The cost of exporting pyruvate into the medium was calculated similarly [30].

$$log\left(\frac{A_o}{A_i}\right)^{eq} = 2(pH_o - pH_i) + \frac{(n-2)(-pmf)F}{ln(10)RT}, \tag{4}$$

$$\left(\frac{A_o}{A_i}\right)^{eq} = \left(\frac{10^{pKa_1+pKa_2-2pH_o} + 10^{pKa_2-pH_o} + 1}{10^{pKa_1+pKa_2-2pH_i} + 10^{pKa_2-pH_i} + 1}\right)\left(\frac{A_o^{2-}}{A_i^{2-}}\right)^{eq}. \tag{5}$$

## 4. Conclusions

The mechanism by which *Rhizopus oryzae* produces fumaric acid has long been debated. By closely varying the medium pH, the urea feed rate and the glucose feed rate, a better understanding of the fumaric acid production has been uncovered. Clear correlations between the point at which ethanol breakthrough occurs and the amount of urea supplied have been found. At a higher urea feed rate, ethanol production begins at a higher glucose feed rate. A drastic difference was found in the metabolism of *R. oryzae* between pH 4 and higher pH values. The lower pH experienced a proportional increase in the yield of fumaric acid to the glucose feed rate. This was attributed to an improved metabolic function. An optimum point for fumaric acid production has been identified. It lies at pH 4 with a urea feed rate of $0.625\,mg\,L^{-1}\,h^{-1}$ and a glucose feed rate of $0.329\,g\,L^{-1}\,h^{-1}$. The yield of fumaric acid at this point is $0.93\,g\,g^{-1}$ with productivity of $0.305\,g\,L^{-1}\,h^{-1}$. This point has total glucose consumption and, therefore, no glucose is wasted or present in the exit stream.

**Author Contributions:** Conceptualization, R.M.S. and W.N.; methodology, R.M. and D.K.R.; software, R.M.S.; validation, R.M.S. and D.K.R.; formal analysis, R.M.S.; investigation, R.M.S. and D.K.R.; resources, W.N.; data curation, R.M.S.; writing—original draft preparation, R.M.S.; writing—review and editing, R.M.S., H.B. and W.N.; visualization, R.M.S.; supervision, W.N. and H.B.; project administration, R.M.S.; funding acquisition, R.M.S. and W.N. All authors have read and agreed to the published version of the manuscript.

**Funding:** This research was funded by the National Research Foundation grant number MND200609529524.

**Data Availability Statement:** The data presented in this study are openly available in the University of Pretoria Research Data Repository at DOI: 10.25403/UPresearchdata.17308847

**Conflicts of Interest:** The authors declare no conflict of interest. The funders had no role in the design of the study; in the collection, analyses, or interpretation of data; in the writing of the manuscript, or in the decision to publish the results.

## Abbreviations

The following abbreviations are used in this manuscript:

| | |
|---|---|
| $A$ | Acid concentration (mol $L^{-1}$) |
| ATP | Adenosine triphosphate |
| $C$ | Carbon mole concentration of specie(Cmol $L^{-1}$) |
| $e$ | Effluent |
| $eq$ | Equilibrium |
| $F$ | Faradays constant (96.5 kJ $Volt^{-1}$e-$mol^{-1}$) |
| $f$ | Feed into reactor |
| $FADH_2$ | Flavin adenine dinucleotide |
| GTP | Guanosine triphosphate |
| $i$ | Intracellular |
| $j$ | Designation of a specie |
| $N$ | Moles of species (Cmol$_i$) |
| $n$ | Number of protons |
| NADH | Nicotinamide adenine dinucleotide |
| $o$ | Extraceluler |
| $pKa$ | Dissociation equilibrium constant |
| $pmf$ | Proton motive force (Volt) |
| P/O | Oxidative phosphorylation value ($mol_{ATP}\ mol_{NADH}^{-1}$) |
| $Q$ | Volumetric (L $h^{-1}$) |
| $R$ | Gas constant ($8.314 \times 10^{-3}$ kJ $mol^{-1}$ $K^{-1}$) |
| $r$ | Rate of production (Cmol$_i$ $L^{-1}$ $h^{-1}$) |
| $T$ | Temperature (K) |
| $t$ | Time increment (s) |
| TCA | Tricarboxylic acid |
| $V$ | Volume of the fermenter (L) |
| $X$ | Biomass |
| $Y_{gx}$ | Yield of biomass on glucose (g $g^{-1}$) |

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
