# Peer review of "Continuous Production of Fumaric Acid with Immobilised Rhizopus oryzae: The Role of pH and Urea Addition"

_catalysts, doi:10.3390/catal12010082_

Round 1
Reviewer 1 Report
The presented research is relevant for industrial biotechnology. However, in this presentation the results do not correspond to the main scopes of the Catalysts. The aim is formulated in a very inconvenient way for the readers. Very scant information is given about the catalytic systems of Rhizopus oryzae involved in the biosynthesis of fumaric acid.
Reviewer 2 Report
Review to the manuscript catalysts-1502596 entitled “Continuous production of fumaric acid with immobilised Rhizopus oryzae: the role of pH and urea addition” by Reuben Marc Swart et al. submitted to Catalysts. The manuscript of a research paper focuses on optimizing continuous production conditions of fumaric acid with immobilised fungal species Rhizopus oryzae.
General remarks:
The subject of the manuscript is in accordance with the aims and scope of Catalysts as it describes an important process for green chemistry. The novelty of the work is not exceptionally high as the fumaric acid production with highly-productive R. oryzae is known for decades, but the application-related research has its value in specifying the conditions to obtain the highest yield. The proposed metabolic flux model can be used in similar studies.
The manuscript is adequately structured and composed, and generally clearly presented but some additions or clarifications have to be made before the manuscript can be considered for publication. The manuscript is written in the correct academic style English language.
Specific comments:
Abstract. It is not mentioned in the abstract that immobilized cells were used to study the process. The fermentation conditions and the used catalyst are very important aspects of the work and should be highlighted. The information on the market size (lines 2-3) is not vital and this information can be presented only in the Introduction. The abstract should be partially modified.
Introduction. The relevant background is given on fumaric acid use in biotechnology and industry and industrially viable processes for fumaric acid production with conditions influencing the product yield. More information and vital details should be provided on the immobilization process and reactor selection for R. oryzae. The statement “Our immobilised R. oryzae process …” should be clarified (lines 40-41). Some specific examples of similarly immobilized microorganisms for dicarboxylic acids production could give wider perspective of the study.
The section (lines 40-46) should be at least partially moved to the end of Introduction as it is summarizing the aim of the study which should appear after the literature overview.
Results and discussion. To facilitate the reading of the manuscript the section should be divided into the subsections with explanatory titles. The Results and discussion section starts very abruptly with “An experimental run consisting… “ (line 73). There should be at least some introductory sentences as the detailed information on the methodology and fermenter design is given below in Materials and Methods.
Materials and methods. The section is generally comprehensive but some of the experimental details are not presented.
Specific comments:
Line 3. There should read Rhizopus oryzae as the species name have not been mentioned previously in the abstract.
Fig. 1a, Fig. 2, Fig. 3, Fig. 4. How many technical parallel measurements were conducted to find average values presented on the figures? It should be indicated in the figure legends.
Lines 132-134. The reference on the pathway should be given.
Lines 250-252. What liquid was used to mix with the spores? What was the final volume of the batch fermentation?
Line 267. Urea is heat sensitive compound. Is it really the case that it was also sterilized by autoclaving?
Lines 270-278. The subchapter is written in the present tense not past tense as the rest of the chapter. The fermenter volume should be mentioned.
Round 2
Reviewer 1 Report
The changes made to the manuscript are quite sufficient.
Reviewer 2 Report
Review to the revised manuscript catalysts-1502596 entitled “Continuous production of fumaric acid with immobilised Rhizopus oryzae: the role of pH and urea addition” by Reuben Marc Swart et al. submitted to Catalysts.
I sincerely thank the authors for their effort. The manuscript readability has been considerably improved during the revision. The issues and comments that were pointed out have been addressed by the authors. The authors dd not include some specific examples of similarly immobilized microorganisms for dicarboxylic acids production to the Introduction to give a wider perspective of the study. As it was only a suggestion, the manuscript could be sufficient without additional information.